# TRAINING-FREE EDITIONING OF TEXT-TO-IMAGE MODELS

## ABSTRACT

Inspired by the software industry's practice of offering different editions or versions of a product tailored to specific user groups or use cases, we propose a novel task, namely, *training-free editioning*, for text-to-image models. Specifically, we aim to create variations of a base text-to-image model without retraining, enabling the model to cater to the diverse needs of different user groups or to offer distinct features and functionalities. To achieve this, we propose that different editions of a given text-to-image model can be formulated as *concept subspaces* in the latent space of its text encoder (*e.g.*, CLIP). In such a concept subspace, all points satisfy a specific user need (*e.g.*, generating images of a cat lying on the grass/ground/falling leaves). Technically, we apply Principal Component Analysis (PCA) to obtain the desired concept subspaces that correspond to specific user needs or requirements from a representative text embedding. Projecting the text embedding of a given prompt into these low-dimensional subspaces enables efficient model editioning without retraining. Intuitively, our proposed editioning paradigm enables a service provider to customize the base model into its "cat edition" (or other editions) that restricts image generation to cats, regardless of the user's prompt (*e.g.*, dogs, people, etc.). This introduces a new dimension for product differentiation, targeted functionality, and pricing strategies, unlocking novel business models for text-to-image generators. Extensive experimental results demonstrate the validity of our approach and its potential to enable a wide range of customized text-to-image model editions across various domains and applications.

## 1 INTRODUCTION

Recent advances in text-to-image models (Zhang et al., 2023b; Rombach et al., 2022; Ramesh et al., 2021; Saharia et al., 2022; Nichol et al., 2021; Betker et al., 2023; Gu et al., 2022) have revolutionized visual content creation, enabling users to create highly realistic images from natural language descriptions. However, as these models become more widely adopted, service providers face challenges in monetizing them and tailoring offerings to diverse customer needs. In the software industry, providers overcome this by offering product editions or versions tailored to specific user segments, *e.g.,* Home Edition, Professional Edition, Enterprise Edition. In this work, we propose a novel task, namely, *training-free editioning* (Fig. 1), and apply this strategy to text-to-image models.

While it may seem straightforward, editioning is a challenging task as it requires preventing users from bypassing access controls. For example, as Fig. 2 shows, the naive solution of *sensitive word filtering* does not work as users can easily evade it using descriptive prompts that are difficult to filter.

Our core idea is creating model variations without retraining to cater to different customer needs or offer distinct features, formulating editions as *concept subspaces* within the embedding space of the model's text encoder. Our concept subspace encapsulates points satisfying a user requirement (*i.e.*, concept), *e.g.*, cat images with specific choices of backgrounds. Technically, we apply Principal Component Analysis (PCA) to text embeddings corresponding to a given concept and retain principal components capturing key variations to obtain a low-dimensional subspace for that concept within the original embedding space. Then, we achieve training-free editioning of text-to-image models by projecting the embeddings of input prompts into these subspaces.

Crucially, our approach allows service providers to efficiently customize the base model into targeted "editions" satisfying diverse customer needs without costly retraining. This leverages the pre-trained

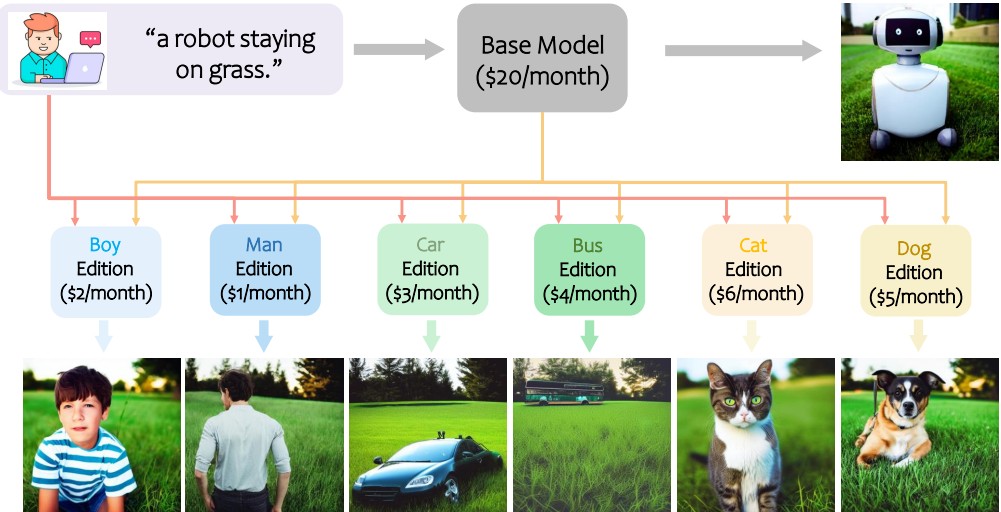

Figure 1: Illustration of **Text-to-Image Model Editioning**. Our method can create variations (e.g., *Boy Edition, Cat Edition*) of a *base* text-to-image model without retraining, enabling them to cater to the diverse needs of different user groups or to offer distinct features and functionalities.

model's capabilities while enabling fine-grained control over outputs. Providers can create tailored editions for different verticals, user types, or functionality tiers - *e.g.* a "cat edition" restricting outputs to cat images regardless of the input prompt (*e.g.*, dogs, people). This unlocks innovative product strategies like freemium models with basic free editions versus feature-rich premium paid editions, enforcing content filters, specialty domains, or custom functionality per edition. Rather than offering an open-ended general tool, our paradigm shifts text-to-image models towards a customizable product portfolio optimized for commercial deployment. Service providers gain flexibility to create an offering tailored to their customer base, introducing novel business models beyond simply vending the base model. This empowers profitably serving diverse market needs while monetizing their AI assets through product differentiation and pricing opportunities better matched to consumer segments. Extensive experiments validate our method's ability to create purposeful model customizations across various domains and applications. Our contributions include:

- We introduce a novel task called "training-free editioning" for text-to-image models, which aims to create customized variations or editions of a base model without expensive retraining.

- We propose a novel method to achieve training-free editioning by formulating different model editions as *concept subspaces* within the text embedding space of the base model, leveraging Principal Component Analysis (PCA) to obtain low-dimensional subspaces capturing desired concepts.

- Extensive experiments across various domains demonstrate the effectiveness of our approach in creating purposeful model customizations suited for different user groups and applications. We highlight the business potential of training-free editioning in enabling service providers to offer differentiated product editions, innovative pricing strategies, and tailored solutions optimized for commercial deployment.

## 2 RELATED WORK

**Text-to-Image Synthesis.** Driven by the success of deep generative models, text-to-image synthesis has become a rapidly evolving field in computer vision and machine learning that aims to generate realistic images from textual descriptions. One of the pioneering works in this field is AlignDRAW (Mansimov et al., 2015), which introduced an attention-based approach that generates images by drawing a sequence of patches on the canvas based on an input caption. While this method represented a promising step forward, the generated images often lacked coherence and failed to accurately reflect the input textual descriptions. The advent of generative adversarial networks (GANs) ushered in a new era of text-to-image synthesis techniques. Text-conditional

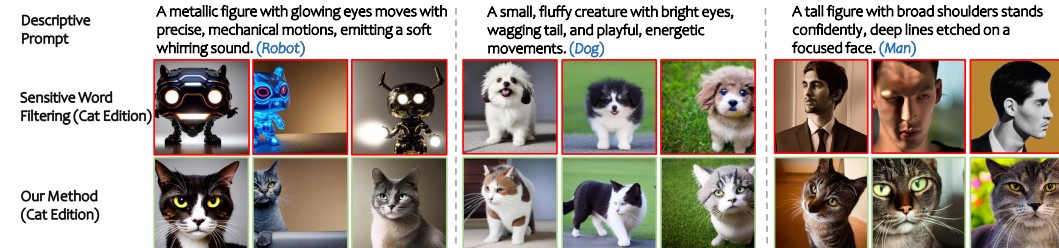

Figure 2: **Sensitive word filtering fails as a naive solution.** Users can easily bypass the access control and generate images beyond the edition (middle) by using descriptive prompts (top) that evade detection by sensitive word filtering methods. In contrast, our method successfully enforces access control (bottom).

GANs (Reed et al., 2016) were among the first to leverage the adversarial training framework for this task. Subsequently, methods like StackGAN (Zhang et al., 2017), AttnGAN (Xu et al., 2018), and ControlGAN (Li et al., 2019) demonstrated improved performance by incorporating attention mechanisms and hierarchical architectures. Despite their notable achievements, these GAN-based approaches often struggled to maintain high consistency, resolution, and diversity in the generated images, falling short of meeting the demanding requirements of real-world applications. A significant breakthrough in text-to-image synthesis emerged with the introduction of large-scale datasets and transformer-based models. OpenAI's DALL-E (Ramesh et al., 2021) pioneered the use of vast text-image pairs, enabling the generation of high-quality images from textual descriptions. Building upon this success, Parti (Yu et al., 2022) further demonstrated the potential of scaling up data and models for improved text-to-image generation performance. Nevertheless, thanks to their stable training and flexible conditioning (e.g., text, image, and other modalities), diffusion models (Rombach et al., 2022) have dominated the state-of-the-art solutions for text-to-image synthesis.

**Diffusion Models.** Diffusion models are a class of deep generative models that have recently demonstrated remarkable performance in generating high-quality samples across various applications. These models are parameterized Markov chains trained using variational inference to generate samples that match the data distribution after a finite number of iterations (Sohl-Dickstein et al., 2015; Ho et al., 2020). Diffusion implicit models (Song et al., 2020), which are based on a class of non-Markovian diffusion processes, lead to the same training objective as traditional diffusion models, but can produce high-quality samples more efficiently. A representative framework for training diffusion models in the latent space is *Stable Diffusion*, a scaled-up version of the Latent Diffusion Model (LDM) (Rombach et al., 2022). Thanks to its flexibility allowing for multi-modal control signals (including text), Stable Diffusion has captivated the imagination of many users and dominated the field, especially in the open-source community. For example, Gal et al. (2022) proposed a novel approach to create variations of a given "concept" by representing it with a single word embedding; Prompt-to-prompt (Hertz et al., 2022) focuses on manipulating the attention maps corresponding to the text embeddings for editing images in pre-trained text-conditioned diffusion models; Null-text inversion (Mokady et al., 2023) proposed performing Denoising Diffusion Image Model (DDIM) inversion on the input image with related prompts into the latent space of a text-guided diffusion model, enabling intuitive text-based image editing. These efforts, while effective, have focused primarily on extending the technical power and usability of the text-to-image (diffusion) model, whose business model is still immature.

To bridge this gap, we propose a novel task called *training-free editioning* for text-to-image models, which aims to create customized variations or editions of a base model without expensive retraining. This enables service providers to offer differentiated product editions, innovative pricing strategies, and tailored solutions optimized for commercial deployment.

## 3 DEFINITION OF TRAINING-FREE EDITIONING

**Definition 1.** Given a trained general-purpose text-to-image model $M$, let $C = \{c_1, c_2, ..., c_n\}$ be a list of $n$ concepts (textual) to be editioned on, $p$ be an input prompt to $M$, we denote the image synthesized by $M$ but editioned on $C$ as:

$$I = M(p \mid C), \tag{1}$$

where $I$ is restricted to only containing concepts in $C$.

## 3.1 Differences with Image Editing and Concept Erasing

**Editioning vs. Editing.** Task-wise, text-to-image editing (Kawar et al., 2023; Rombach et al., 2022; Hertz et al., 2022; Brooks et al., 2023; Mokady et al., 2023; Tumanyan et al., 2023; Yang et al., 2023; Couairon et al., 2022) works at the *aspect/image-level*, which typically refers to the process of modifying or manipulating specific aspects of an existing image based on a text prompt or instructions while leaving irrelevant aspects of it unchanged, e.g., inpainting, outpainting, or style transfer. In contrast, the proposed text-to-image editioning task works at the *model-level*, which aims to customize the behavior of the text-to-image model itself to cater to specific user needs or functionalities. Methodology-wise, since editing works at the *aspect/image-level*, its key challenge is to disentangle the target aspect of an image that needs to be edited from the rest, e.g., by manipulating the attention maps of a given image (Xu et al., 2023; Ju et al., 2023; Li et al., 2023; Hertz et al., 2022; Mokady et al., 2023). On the contrary, our editioning task is performed at the *model level*, so its main challenge lies in controlling the model's behavior, e.g., by manipulating the model's text/image embedding space.

**Editioning vs. Concept Erasing.** Task-wise, our editioning and concept erasing can be viewed as *complementary tasks*, where our editioning aims to retain concept(s) $C$ from a model and concept erasing aims to remove $C$ from the model. Methodology-wise, existing concept erasing methods (Gandikota et al., 2023; Kumari et al., 2023; Gandikota et al., 2024; Liu et al., 2023; Huang et al., 2023; Yildirim et al., 2023; Zhang et al., 2023a; Kim et al., 2023) primarily focus on fine-tuning model weights. In contrast, our approach does not involve any training and concentrates on manipulating the model's text/image embedding space directly. The choice of such distinct methodologies stems from the observation that $C$ typically constitutes a relatively small subset compared to the entire set of concepts learned by the model. Consequently, for concept erasing, removing $C$ can be achieved through a minor perturbation of the model weights. However, for our editioning task, retaining $C$ and dropping all other concepts would necessitate a significant modification, akin to retraining the entire model from scratch.

## 4 Method

Addressing Definition 1, we propose a novel approach, namely **Concept Subspace Projection**, which achieves $M(p \mid C)$ by projecting the embedding vector of $p$ to a concept subspace $S_{\mathcal{E}}(C)$ defined by $C$. Specifically, let $\mathcal{E}$ and $\mathcal{G}$ be the text encoder and generator of $M$, respectively, i.e., $M(\cdot) = \mathcal{G}(\mathcal{E}(\cdot))$, $S_{\mathcal{E}} = \mathbb{R}^d$ be the $d$-dimensional embedding space of $\mathcal{E}$, we have:

$$\mathcal{E}(p \mid C) = PR_{S_{\mathcal{E}}(C)}(\mathcal{E}(p)) \tag{2}$$

where $PR_x(y)$ denotes the projection of $y$ on $x$, $S_{\mathcal{E}}(C) = \mathbb{R}^{d_C} \subset S_{\mathcal{E}}$ denotes the $d_C$-dimensional concept subspace specified by concepts $C$, $d_C < d$. In this way, we create the **C-edition** of $M$ as the concept space $S_{\mathcal{E}}(C)$ and have:

$$I = M(p \mid C) = \mathcal{G}(\mathcal{E}(p \mid C)) = \mathcal{G}(PR_{S_{\mathcal{E}}(C)}(\mathcal{E}(p))) \tag{3}$$

### 4.1 CLIP-based Concept Subspace Projection

Recognizing that CLIP (Radford et al., 2021) dominates the implementation of $\mathcal{E}$ in state-of-the-art text-to-image models (Zhang et al., 2023b; Rombach et al., 2022; Ramesh et al., 2021; Saharia et al., 2022; Nichol et al., 2021; Betker et al., 2023; Gu et al., 2022), we follow this common practice and develop our method in the CLIP embedding space. Thanks to CLIP's use of **cosine similarity** for comparing text and image embeddings, we hypothesize that (please see Sec. 5.3 for an empirical justification):

**Conjecture 1.** For CLIP encoders, the text embeddings in concept subspaces $S_{\mathcal{E}}(C)$ corresponding to different $C$ are on a thin **hypersphere shell centered at the origin**.

**Concept Subspace Creation.** Based on Conjecture 1, the concept subspace $S_{\mathcal{E}}(C)$ accommodating the hypersphere shell can be characterized by a set of (orthogonal) vectors radiating from the origin. To obtain such vectors, we propose applying Principal Component Analysis (PCA) to a substantial sample of $\mathcal{E}(p_C)$ embeddings $D_C = [\mathcal{E}(p_C^1), \mathcal{E}(p_C^2), ..., \mathcal{E}(p_C^m)]$:

$$\mathbf{V}_C = \text{PCA}(D_C) \tag{4}$$

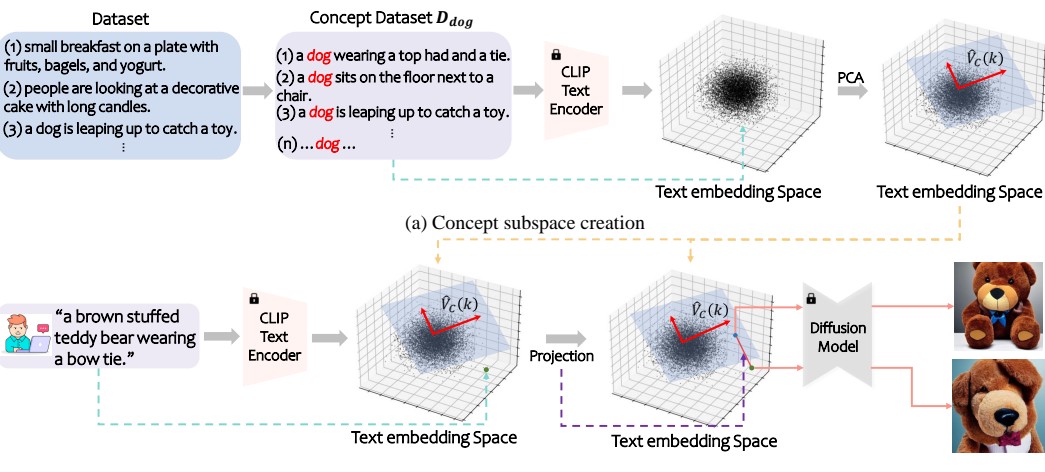

(a) Concept subspace creation

(b) Concept subspace projection

Figure 3: Overview of our concept subspace creation (top) and projection (bottom).

where $p_C$ denotes a prompt that contains only the concepts in $C$, $\mathbf{V}_C$ denotes the principal axes ranked by descending principal values. Then, we define:

$$\hat{\mathbf{V}}_C(k) = \mathbf{V}_C[0:k] \tag{5}$$

as the basis of subspace $S_{\mathcal{E}}(C)$, where $k$ is selected according to a 95% threshold of explained variance. Please see Sec. 5.4 for more details.

**Magnitude-compensated Projection.** With $\hat{\mathbf{V}}_c(k)$, we define the projection function $PR$ as:

$$PR_{S_{\mathcal{E}}(C)}(\mathcal{E}(p)) = \eta \cdot \hat{\mathbf{V}}_c(k) \cdot \hat{\mathbf{V}}_c(k)^T \cdot \mathcal{E}(p) \tag{6}$$

where $\eta = \frac{||\mathcal{E}(p)||}{||\hat{\mathbf{V}}_c(k) \cdot \hat{\mathbf{V}}_c(k)^T \cdot \mathcal{E}(p)||}$ is a parameter to compensate for the loss of magnitude during the projection. Note that we omit the centering step for simplicity, since the PCA subspace is also approximately centered at the origin (Conjecture 1).

**Efficient Computation.** Guided by the classic manifold hypothesis (Brown et al., 2022) that assumes the existence of low-dimensional representations of high-dimensional data, we apply PCA to a substantial random sample of CLIP text embeddings to reduce the dimensionality of the original CLIP embedding space from $77 \times 768 = 59,136$ to $13,000$. This "compression" significantly improves computational efficiency by roughly $(59,136/13,000)^2 \approx 20.7$ times for the computation of covariance matrix in PCA (bottleneck), without sacrificing the performance (Fig. 4). Thus, unless specified otherwise, we employ the 13,000-dimensional reduced space as the CLIP text embedding space in our experiments.

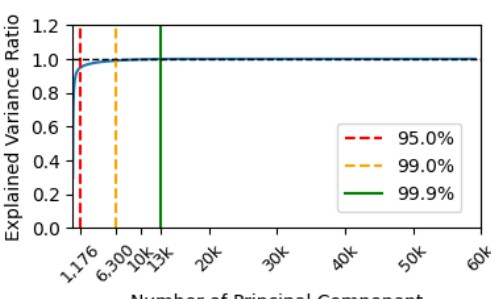

Figure 4: 13k components yield a cumulative explained variance ratio of 99.9+%.

## 5 EXPERIMENTS

### 5.1 EXPERIMENTAL SETUP

As mentioned above, our method consists of two steps: i) performing a low-loss dimensionality reduction to obtain an "efficient" CLIP embedding space of 13,000 dimensions; ii) creating concept spaces and projecting the embeddings of input prompts to them using the method proposed in Sec. 4. To implement them, we created the following datasets:

Table 1: CLIP score (softmax probability) of the images generated by our concept subspace projection, and their corresponding "ground truth" prompts (i.e., those accurately describing the image content).

| Concept Subspace | Animal | | | Vehicle | | | Human | | |
|---|---|---|---|---|---|---|---|---|---|
| | Dog | Cat | Tiger | Car | Bus | Truck | Boy | Girl | Man |
| Clip Score | 0.9594±0.1702 | 0.9105±0.2304 | 0.8803±0.2556 | 0.9020±0.2473 | 0.8943±0.2508 | 0.9270±0.1905 | 0.8953±0.2484 | 0.8543±0.2906 | 0.8808±0.2646 |

Table 2: Evaluating the image synthesis capability of our concept space projection method using FID and IS scores. Ours: for each concept subspace, we take its evaluation dataset and generate their corresponding 4,000 images using the proposed concept space projection; SD: for each concept subspace, we replace the subject of the prompts used in "Ours" with the concept of the subspace and generate 4,000 images using Stable Diffusion v1.4 (Rombach et al., 2022)). SD': different sets of 4,000 images generated in the same way as "SD".

| | | Dog | Cat | Tiger | Car | Bus | Truck | Boy | Girl | Man | Mean |
|---|---|---|---|---|---|---|---|---|---|---|---|
| FID | Ours vs. SD | 14.982 | 38.236 | 34.845 | 14.317 | 32.921 | 21.789 | 14.350 | 14.215 | 16.125 | 20.405 |
| | SD vs. SD' | 6.723 | 6.143 | 2.390 | 6.239 | 3.093 | 3.887 | 9.977 | 10.035 | 11.081 | 6.619 |
| IS | Ours | 10.150 | 4.012 | 2.754 | 5.820 | 4.102 | 3.850 | 9.500 | 9.550 | 11.300 | 6.870 |
| | SD | 8.600 | 2.600 | 1.200 | 4.400 | 2.250 | 2.000 | 8.800 | 8.900 | 10.600 | 5.600 |

**CLIP Dimensionality Reduction Dataset** ($D_{all}$)**.** CoCo 2017 Dataset (TY Lin) contains thousands of image and caption pairs. We randomly selected a subset of 160,000 captions from it and embedded them with CLIP to create the dataset $D_{all}$ for the dimensionality reduction in creating the 13,000-dimension "efficient" CLIP embedding space.

**Concept Datasets.** Since *subjects* are usually of the most interest to users performing text-to-image synthesis and editing tasks, without loss of generality, we focus on the concepts of *subjects* in our experiments. Therefore, except $D_{all}$ mentioned above, we create our concept datasets by extracting all captions in the CoCo 2017 dataset that contain certain subjects (e.g., $D_{cat}$ is the union of all CoCo 2017 captions containing "cat" as their subjects). Note that we remove captions with pronouns (e.g., 'that', 'this') as *subjects* as they have no specific meanings. We created 9 such datasets in our experiments, including i) Animals: $D_{cat}$, $D_{dog}$, and $D_{tiger}$; ii) Vehicles: $D_{car}$, $D_{bus}$, and $D_{truck}$; iii) Human: $D_{boy}$, $D_{girl}$, and $D_{man}$.

**Concept Subspaces Creation and Evaluation.** We follow the method detailed in Sec. 4.1 to create our concept subspaces, e.g., $S_{\mathcal{E}}(\text{cat})$ and $S_{\mathcal{E}}(\text{dog})$, using their corresponding concept datasets, e.g., $D_{cat}$ and $D_{dog}$, respectively. In addition, given a concept subspace $S_{\mathcal{E}}(*)$, we construct its evaluation dataset by randomly selecting 1,000 captions from $D_{all}$ whose subjects are not $*$.

**Evaluation Metrics.** We use i) CLIP scores (Hessel et al., 2021) to measure the consistency between an image generated by our method and its corresponding prompt (before and after our content subspace projection); ii) Fréchet Inception Distance (FID) (Heusel et al., 2017) and Inception Score (IS) to measure the image synthesis ability of the base model and its editions created by our method.

## 5.2 Effectiveness of Concept Subspace Projection for Text-to-Image Model Editioning

### 5.2.1 Quantitative Results

**Editioning Accuracy.** We use the CLIP score (probability) (Hessel et al., 2021) to measure the editioning accuracy of our method, with 0 indicating low accuracy and 1 indicating high accuracy. Specifically, given a concept space $S$ (e.g., cat edition $S_{\mathcal{E}}(\text{cat})$) created using $D$ (e.g., $D_{cat}$) defined in Sec. 5.1, for each input prompt $p$ in its evaluation dataset, we compute the softmax probability of: i) the CLIP score between the images $I$ generated using the projected prompts and their corresponding "ground truth" prompts $\hat{p}$, i.e., replacing the corresponding concept in $p$ with that of $D$ (e.g., in the cat edition $S_{\mathcal{E}}(\text{cat})$, for the randomly selected prompt, the subject concept can be exchange into "cat"); ii) the CLIP score between $I$ and $p$. Since the sum of the two probabilities is 1, we report the former in Table 1, which demonstrates that our concept subspace projection accurately restricts the generation to the concept of $S$ (all scores are high).

Table 3: Cosine similarities between the input prompt, its projected version, and its "replaced" version. The "replaced" version refers to the text embedding of the prompt created by replacing the subject component in the input prompt (e.g., "dog") with that of the concept word in the concept subspace (e.g., "cat") being projected onto. The input prompts used are from the corresponding evaluation dataset.

| | $d(\text{input}, \text{replace})$ | $d(\text{project}, \text{replace})$ | | $d(\text{input}, \text{replace})$ | $d(\text{project}, \text{replace})$ |
|---|---|---|---|---|---|
| $S_{\mathcal{E}}(\text{dog})$ | $0.1985\pm0.0687$ | $0.1674\pm0.0601$ | $S_{\mathcal{E}}(\text{truck})$ | $0.2376\pm0.0728$ | $0.1975\pm0.0497$ |
| $S_{\mathcal{E}}(\text{cat})$ | $0.2114\pm0.0683$ | $0.1785\pm0.0611$ | $S_{\mathcal{E}}(\text{boy})$ | $0.1899\pm0.0792$ | $0.1733\pm0.0796$ |
| $S_{\mathcal{E}}(\text{tiger})$ | $0.2384\pm0.0743$ | $0.2197\pm0.0540$ | $S_{\mathcal{E}}(\text{girl})$ | $0.1953\pm0.0785$ | $0.1751\pm0.0603$ |
| $S_{\mathcal{E}}(\text{bus})$ | $0.2525\pm0.0796$ | $0.2693\pm0.0676$ | $S_{\mathcal{E}}(\text{man})$ | $0.1782\pm0.0738$ | $0.1963\pm0.0742$ |
| $S_{\mathcal{E}}(\text{car})$ | $0.2172\pm0.0713$ | $0.1623\pm0.0456$ | Mean | $0.2132$ | $0.1932$ |

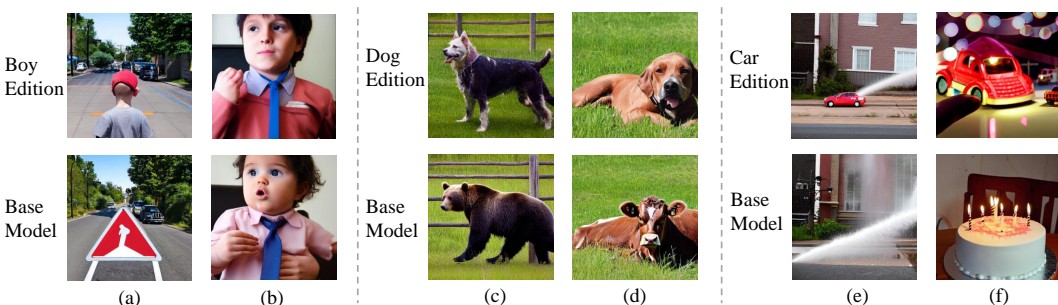

Figure 5: Images generated by different prompts when using different editions of the Stable Diffusion v1.4 model. The input prompts are: (a) a street sign reading give way next to a road. (b) a baby plays with an adult-sized tie put on him. (c) a bear walks along a fence on a plain. (d) a brown cow lays in the grass on a hill. (e) the fire hydrant is shooting water into the street. (f) a birthday cake replicates a demolition scene with candles.

**Image Synthesis Capability.** We use the FID and IS scores to measure the image synthesis capability, with reference to those of the base model, i.e., Stable Diffusion (SD) v1.4 (Rombach et al., 2022). As Table 2 shows, the FID scores of our method are worse than those of SD but our IS scores are higher, indicating that our method generates similarly high quality but less diverse images than SD.

**Similarity between Text Embeddings.** To further characterize our content subspace projection, we compute the cosine similarities between the input prompt, its projected version, and its "replaced" version, where the "replaced" version refers to the text embedding of the prompt created by replacing the subject component in the input prompt (e.g., "dog") with that the concept word of the concept subspace (e.g., "cat") being projected onto. As Table 3 shows, the projected embeddings maintain similar distances to the "replaced" ones as input prompts, suggesting that our method operates throught a different mechanism than naive replacement.

### 5.2.2 QUALITATIVE RESULTS

**Editioning Accuracy.** As Fig. 5 shows, we achieved a high editioning accuracy as the target concept (i.e., subject) of the input prompt is restricted to the concept subspace while other concepts (e.g., behavior and background) remain unchanged.

**Image Synthesis Capability.** As Fig. 6 shows, the images generated using our concept subspace projection are of similarly high quality and diversity to those generated by the base model directly.

### 5.3 PROPERTIES OF CLIP CONCEPT SUBSPACES

**Distance to Origin.** As Fig. 7 shows, we plot the histogram of the distances of text embeddings to the origin for each concept dataset and the Coco 2017 dataset. It can be observed that for all datasets, the distances are around 250 with small standard deviations, which justifies our Conjecture 1.

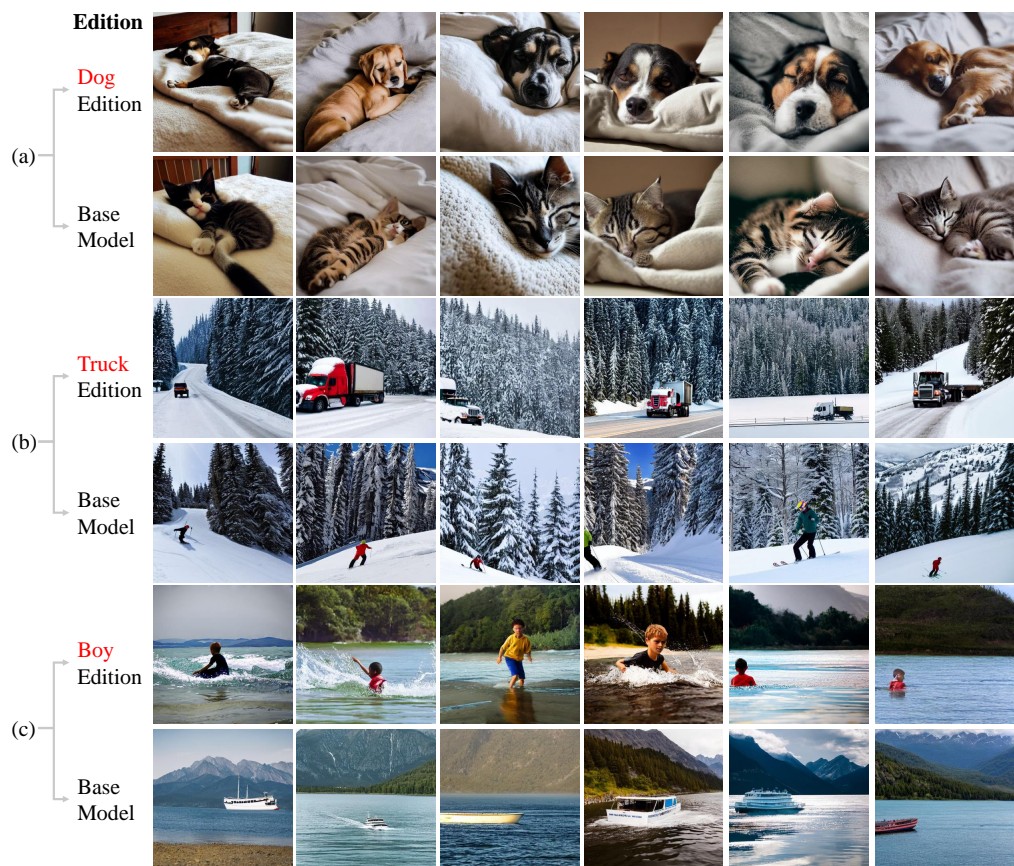

Figure 6: Different images generated by the same prompt when using different editions of the Stable Diffusion v1.4 model. The input prompts are: (a) a kitty all cozy sleeping on a bed. (b) a skier moves down the slope with trees in the background. (c) a boat travels through the water near the mountains.

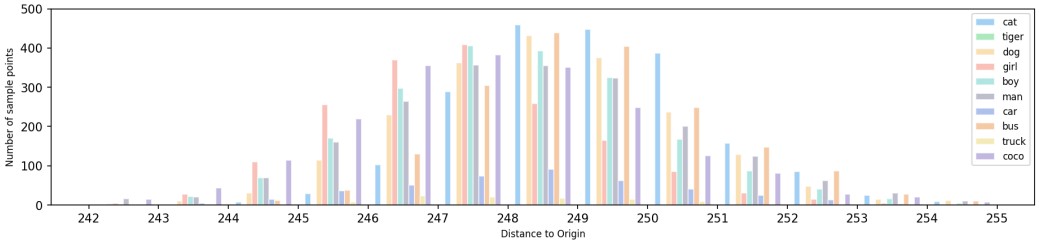

Figure 7: Distances of text embeddings to the origin. We randomly selected 2,000 prompts from each concept dataset and the Coco 2017 dataset to calculate the distances of samples to the origin. The mean and standard deviation values of the distances are shown in the legend.

**Semantic Directions in Concept Subspace.** As Fig. 8 shows, we observed that the principal components of each concept subspace also have semantic meanings. In addition, the content of the image remains restricted to its corresponding conceptual subspace (edition).

**Concept Subspace Interpolation.** As Fig. 9 shows, our concept subspace also allows for linear interpolation between projected text embeddings.

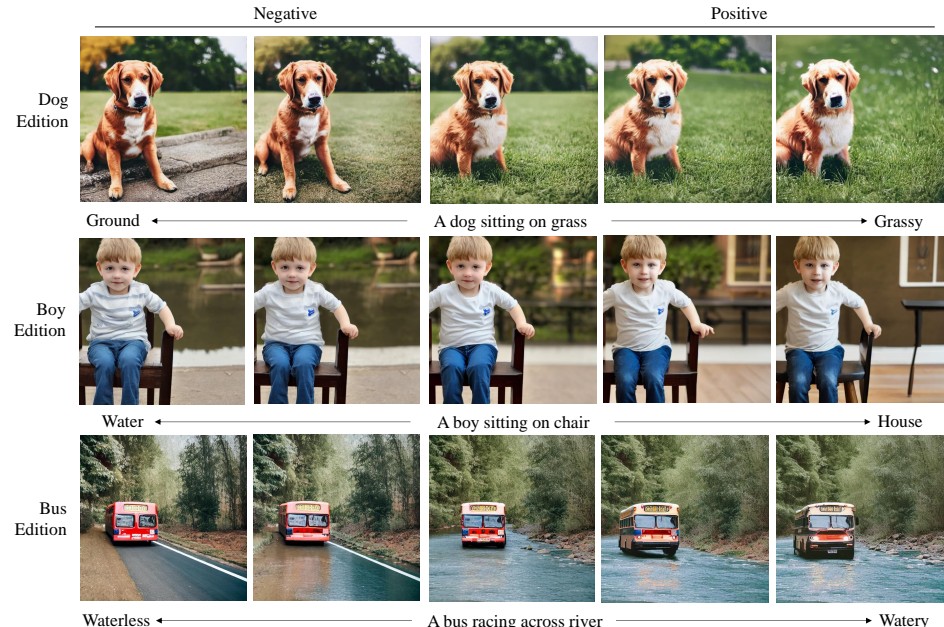

Figure 8: Images generated by moving text embeddings (the input prompt is shown in the middle) along the directions of principal components (PC). Row 1: PC #7; Row 2: PC #14; Row 3: PC #0.

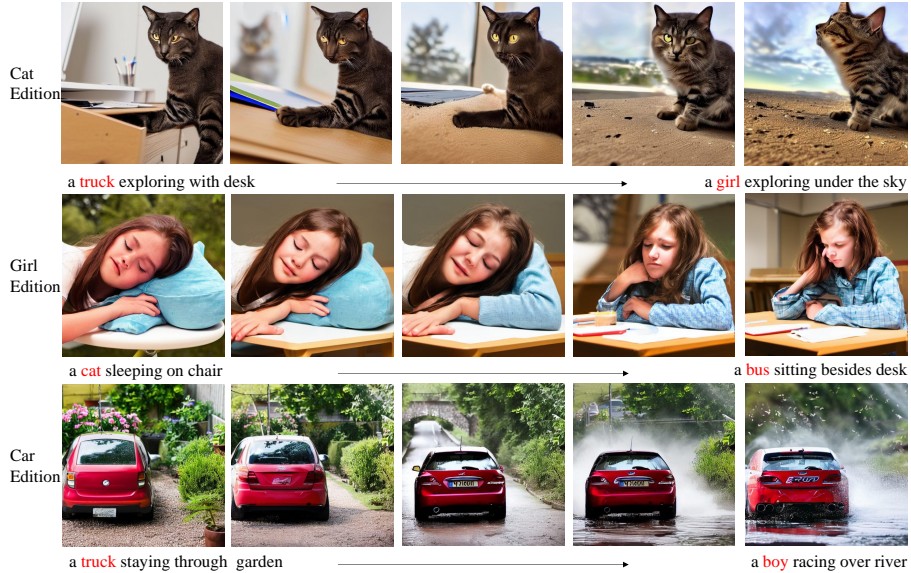

Figure 9: Linear interpolation between projected text embeddings. The input prompts are shown at the bottom. The words in red denote the subject to be restricted to the edition given on the left.

Table 4: Choice of $k$ (Eq. 5) by 99% explained variance ratio for each concept subspace.

| | Dog | Cat | Tiger | Car | Bus | Truck | Boy | Girl | Man |
|---|---|---|---|---|---|---|---|---|---|
| # of Principal Components | 44 | 39 | 23 | 43 | 15 | 42 | 49 | 62 | 64 |

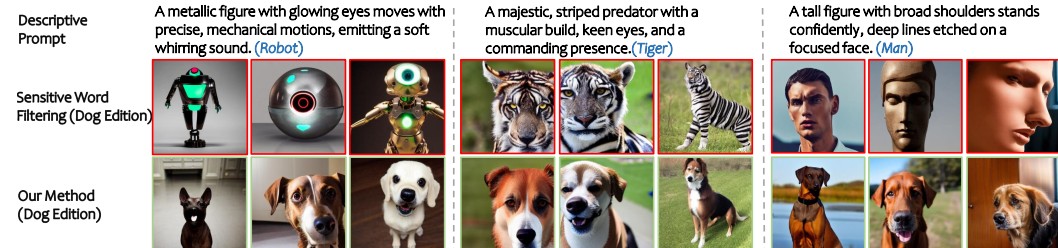

Figure 10: **Sensitive word filtering fails as a naive solution.** Users can easily bypass the access control and generate images beyond the edition (middle) by using descriptive prompts (top) that evade detection by sensitive word filtering methods. In contrast, our method successfully enforces access control (bottom).

Table 5: Computational costs of our method. E.S.: Embedding Space; $(\cdot)d$: $(\cdot)$ dimension.

| Original CLIP E.S $(59,136d)$ | Our Reduced E.S $(13,000d)$ |
|---|---|
| 7h 23min 7s $\pm$ 23s | 1min 11s $\pm$ 12s |

(a) Concept subspace creation

| Text Embedding | Concept Space Projection | Diffusion Generation |
|---|---|---|
| 72ms $\pm$ 34ms | 21ms $\pm$ 8ms | 5s 322ms $\pm$ 1s 483ms |

(b) Image generation (inference)

### 5.4 Choice of $k$ for Each Concept Subspace

As Table 4 shows, empirically, we choose $k$ (Eq. 5) for each concept subspace by the threshold of 99% explained variance ratio of that subspace.

### 5.5 Sensitive Word Filtering Fails as a Naive Solution

To further justify the motivation and usefulness of our approach, in Fig. 2 and Fig. 10, we show that the naive solution of *sensitive word filtering* fails to enforce effective access control and can be easily bypassed by users, whereas our method successfully prevents such bypasses.

### 5.6 Computational Costs

As shown in Table 5, our method is highly efficient, with its time cost (21ms) being negligible compared to the image synthesis time (around 5 seconds) of Stable Diffusion. Moreover, the dimensionality reduction employed by our method (i.e., from 59,136 to 13,000 dimensions) significantly reduces the time required to create a concept subspace from around 7 hours to around 1 minutes.

All experiments in our work are conducted on a workstation with an 12th-gen Intel Core i7-12700 CPU, an Nvidia RTX 4090 24G GPU, 64GB memory and a 1TB hard disk.

## 6 Conclusion

Inspired by software editioning, we propose training-free "editioning" of text-to-image models by identifying *concept subspaces* within the latent space of their text encoders (*e.g.*, CLIP). These subspaces, obtained via applying Principal Component Analysis (PCA) on representative text embeddings, correspond to specific concepts like "cat", "dog", "boy". Projecting the text embedding of a given prompt into these low-dimensional subspaces enables efficient model customization without retraining. This unlocks novel business models, such as offering restricted "cat editions" that only generate cat images regardless of subjects in input prompts, enabling new product differentiation and pricing strategies.

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

## A    RESULTS ON STABLE DIFFUSION v1.5

As shown in Table 6 and Fig. 11, our method also generalizes to Stable Diffusion v1.5 and achieves similarly high CLIP scores and editioning accuracy.

Table 6: CLIP score (softmax probability) of the images generated by our concept subspace projection, and their corresponding "ground truth" prompts (i.e., those accurately describing the image content).

| Concept Subspace | Animal | | | Vehicle | | | Human | | |
|---|---|---|---|---|---|---|---|---|---|
| | Dog | Cat | Tiger | Car | Bus | Truck | Boy | Girl | Man |
| Clip Score | 0.9275±0.1794 | 0.9019±0.2510 | 0.8939±0.2645 | 0.9104±0.2834 | 0.8893±0.2632 | 0.8882±0.2619 | 0.8916±0.2816 | 0.8603±0.2903 | 0.8819±0.2910 |

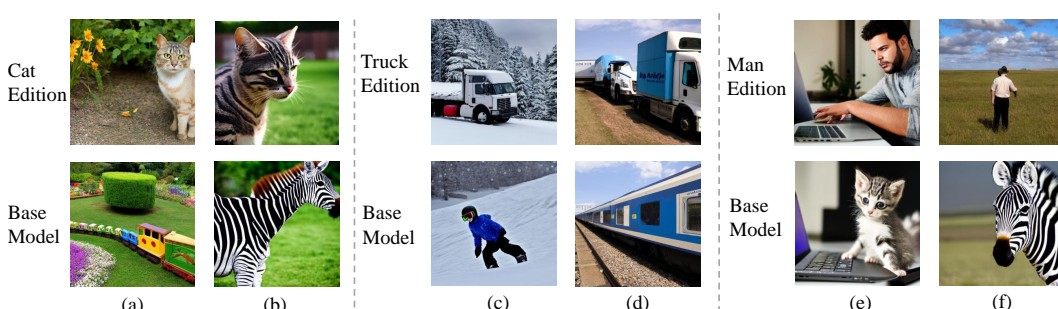

(a)                     (b)                     (c)                     (d)                     (e)                     (f)

Figure 11: Images generated by different prompts when using different editions of the Stable Diffusion v1.5 model. The input prompts are: (a) a mini train travels through a large garden. (b) a zebra stands in his habitat in captivity. (c) a child snowboarding down a hill in the snow. (d) a row of blue and white train cars. (e) a kitten sits facing an open black laptop. (f) a zebra that is standing in a field.

## B    CONCEPT SUBSPACES OF "VERB" AND "OBJECT"

Following a similar experimental setup used for "subject" in the main paper, we show that the proposed method can also be applied to "verb" and "object". As shown in Table 7, Fig. 12, Fig. 13, our method can also accurately restrict the generation to the concept subspace.

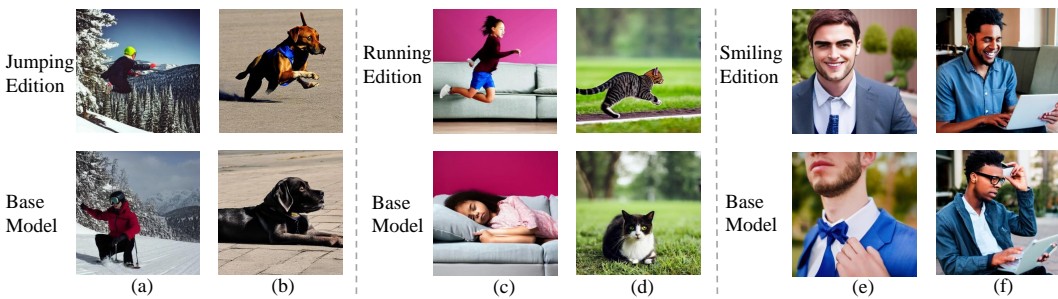

(a)                     (b)                     (c)                     (d)                     (e)                     (f)

Figure 12: Concept Subspaces of $< \text{verb} >$. Images generated by different prompts when using different editions of the Stable Diffusion v1.4 model. The left clarifies the different editions of the object and the base model. The input prompts are: (a) a person crouches low to ski over snowy ground. (b) a dog lying on the ground at sunny day. (c) the girl is sleeping on the sofa. (d) cat stays on the grass with a tree behind it. (e) a young man in a blue shirt admires his tie. (f) a young man stops to look at his electronic device.

Table 7: CLIP score (softmax probability) of the images generated by our concept subspace projection, and their corresponding "ground truth" prompts (i.e., those accurately describing the image content).

| Concept Subspace | Verb | | | Objective | | |
|---|---|---|---|---|---|---|
| | jumping | running | smiling | Table | Grass | Leaves |
| Clip Score | 0.8494±0.2232 | 0.8843 0.2142 | 0.8196±0.2737 | 0.8299±0.2978 | 0.8929±0.2596 | 0.8558±0.2885 |

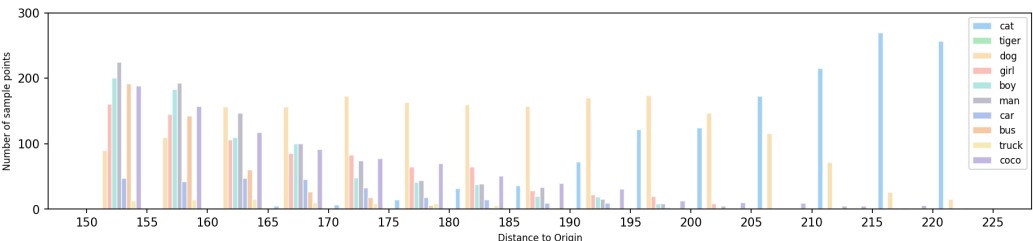

Figure 13: Concept Subspaces of $< \text{object} >$. Images generated by different prompts when using different editions of the Stable Diffusion v1.4 model. The left clarifies the different editions of the object and the base model. The input prompts are: (a) a car drives through on the road. (b) a dog lying on the ground at sunny day. (c) a boy in shirt flying a kite on beach. (d) a brown bear walks lazily along the dirt. (e) a cat lazily lay on the table. (f) the boy wearing a blue sweater sleeping on the chair.

## C    EFFECTIVENESS OF OUR MAGNITUDE-COMPENSATED PROJECTION

**Distances of Text Embeddings to the Origin after Naive Projection.** To demonstrate the necessity of our magnitude-compensated projection (Eq. 6), we show that the distances indeed reduce after naive projections (Fig. 14).

**Qualitative Comparison.** As Fig. 15 shows, without our magnitude-compensated projection (i.e., naive projection), the generated images suffer from severe distortions, which further demonstrates the effectiveness of our magnitude-compensated projection.

Figure 14: Distances of text embeddings to the origin after naive projection. We randomly selected 2,000 prompts from the evaluation dataset of a given concept space $S$ and naively projected them to $S$. The mean and standard deviation values of the distances are shown in the legend.

## D    LIMITATIONS

Our work is a first step toward the new task of "Training-free Editing of Text-to-image Models". As such, it is constrained by the number of concepts in the editions. Nonetheless, we believe that our approach is a solid step forward and will inspire the community for subsequent innovations.

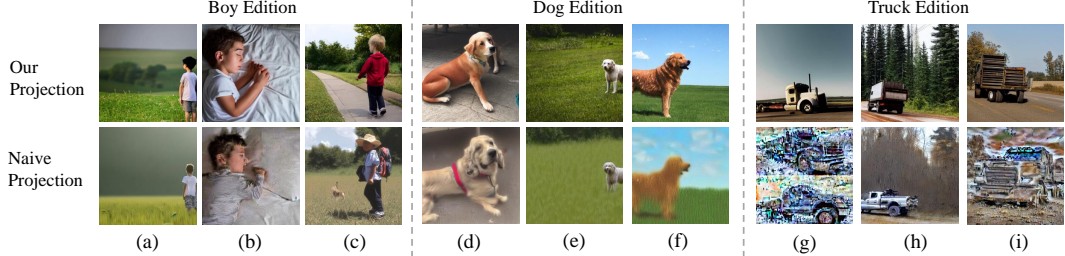

Figure 15: Comparison of images generated using naive projection and our magnitude-compensated projection. All images are generated with editions of the Stable Diffusion v1.4 model. The editions are shown at the top. The input prompts are: (a) a light colored bull stands in a field. (b) a kitty all cozy sleeping on a bed. (c) a horse gazes into the distance. (d) a cat sits at the ready in a mostly empty station. (e) a light colored bull stands in a field. (f) a bear gazes into the sky. (g) a horse running on the dirt path. (h) a bear walks down a trail in the forest. (i) a car stops in the middle of the road.

