# OpenReview forum: "Training-free Editioning of Text-to-Image Models"
_ICLR.cc/2025/Conference — Submitted to ICLR 2025_

### Official Review · Reviewer_UdHw · 2024-10-30

**Soundness:** 1
**Presentation:** 2
**Contribution:** 1
**Rating:** 3
**Confidence:** 5

**Summary:**

This paper focus on the "training-free editioning" task based on a pre-trained text-to-image model. To achiever the objective, the paper proposes some aligned concepts for the input prompts, and apply the PCA method to obtain the desired subspaces where the input and based concepts can be consist with each other. The paper announced that they introduce the novel task.

**Strengths:**

This paper attempts to replace the concept with the PCA model in the subspace of the label, and implements it as training-free editing.

**Weaknesses:**

1. The task proposed in this paper is one of the most basic tasks of text-to-image task.
2. A lot of works have done to align the subjects during the generating process.
3. There is no comparative experiments with other outstanding models, like SDXL, FLUX 1.0.
4. The related work part just list some older papers and there is  insufficient relevance to the question studied in this paper.
5. In the method part, There is only a simple introduction to the PCA method in the 4.1 CLIP-BASED CONCEPT SUBSPACE PROJECTION. No more subject follow it.
6. In the experiment part, the 5.2.2 and 5.3 just list the general words without any in-depth analysis.

**Questions:**

1. Please provide more results with better pre-trained models, like SDXL.
2. What’s the advantage of PAC in this paper compared with LLMs for text alignment.
3. Provides more research papers which are related to the prompt processing and are published in 2024.

---

### Official Review · Reviewer_rvbP · 2024-11-03

**Soundness:** 3
**Presentation:** 3
**Contribution:** 2
**Rating:** 5
**Confidence:** 5

**Summary:**

In this study, the authors propose a new task, ''training-free editioning'', for business T2I services. The goal of the new task is to force the base T2I model to only generate images from a predefined concept set. To avoid retraining the base model, the authors project the input textual prompts to the given concepts. Experiments show the method would be a valid solution for the proposed new task.

**Strengths:**

Overall, the reviewer appreciates the efforts in defining a new task for T2I applications, which genuinely will shed some new light on the research field. Despite much research that has tried to generate/find desired textual prompts for different goals, this is the first time the reviewer has considered controlling the output of T2I models by projecting the textual prompts. The proposed method, though somewhat straightforward, is reasonable. The discussions on Sec.3.1 are also welcomed.

**Weaknesses:**

1. Limited scope. Though the authors have tried their best to show the potential applications for the newly proposed task, the reviewer is not convinced that the proposed solution will be widely used in the T2I community. In what situation could we need to force a T2I service to generate only "cat" images while "must not" other concepts? For concept erasing or model aligning, the goal is clear and easy to achieve by listing unwanted concepts. Then, most of the common concepts would work as usual. In the proposed editioning setting, the outputs can only be one concept (which could also be an unsafe concept if the service wants).
2. The proposed PCA-based method is niche. When reading the definition in Section 3, the reviewer thought the concepts in C could be different objects or attributes. However, as shown in the experiments, one edition of T2I model could only support one kind of subject. Related to the first issue, such a setting is also nearly impractical.
3. Technical writing should be given more attention. The reviewer found it hard to locate the manuscript in related works. The new task confused the reviewer when reading most parts of the introduction. Figure 1 is not helpful. The second paragraph comes in a strange way. The authors are suggested to re-phase the introduction to clearly show the context of this manuscript.
4. The reviewer is surprised to find no user study evaluation is provided in the manuscript.

**Questions:**

Q1. How will the number of prompts in C influence the final performance?
Q2. How could the proposed method deal with multiple concepts? Besides subjects, how does it work on attributes?

---

### Official Review · Reviewer_ZfZa · 2024-11-03

**Soundness:** 3
**Presentation:** 3
**Contribution:** 3
**Rating:** 5
**Confidence:** 4

**Summary:**

This paper proposes a training-free image editioning method aimed for providing controllable image generation for specific user groups or uses cases. They apply PCA to obtain the desired concept subspaces and projects the text embedding of a given prompt into these desired concept low-dimensional subspaces enables efficient model editioning. Extensive experiments demonstrate the effectiveness of the proposed methods.

**Strengths:**

a.The proposed method is simple but effective. The author propose a novel task for offering different editions or versions of a product tailored to specific user groups or use cases, which may be useful in the software industry’s practice.

b.The author conducts extensive experiments and proves its effectiveness.

**Weaknesses:**

a. The proposed method, while eliminating the need for model retraining, still requires the creation of a desired concept dataset in advance, which somewhat restricts its applications.

b. The presentation of method part is not very clear, especially Figure 3 and I didn’t find its citation in the main text.

c.What if I need some concepts that are rare or even not included in the pre-prepared dataset?

d.What if  I want to generate images with multiple concepts rather than a single concept?

e. The pretrained model used in this paper is sd 1.4, what about its effectiveness on other diffusion models?

**Questions:**

See weaknesses.

---

### Official Review · Reviewer_EyFk · 2024-11-05

**Soundness:** 2
**Presentation:** 2
**Contribution:** 2
**Rating:** 5
**Confidence:** 4

**Summary:**

This paper proposes a new task named image editioning that only allows editing a specific concept without training the text-to-image model. To achieve this, this paper projects the CLIP text embedding with PCA into concept subspaces and uses the principal axes corresponding to the concept for editing. The resulting system could only generate images of the desired concept regardless of subjects in input prompts. Experiments demonstrate that the proposed method can achieve the proper results on this new proposed task.

**Strengths:**

- This paper aims to introduce a new type of task that ignores subjects in the text prompts while only editing a predefined concept (e.g., cat).

- This paper analyzes the PCA space in CLIP text encoder embedding and achieves image editioning.

- Several experiments and interesting observations are made for this new task and the proposed method.

**Weaknesses:**

- The image editioning setup is a bit weird to me. If I would like to only generate specific objects, why not just replace the original object's name with the desired concept? An LLM could easily be used for that purpose. The practical value of the proposed method should be better explained.

- Different editions require the same computations, and I do not see why different editions should charge differently as illustrated in Figure 1. I have a hard time figuring out why this task could stimulate new business models.

- What if the prompt contains multiple objects? Can this method select the correct one? Could the proposed method be extended to edit multiple objects/concepts? These points need some discussion.

- Many related works have been done for editing the styleGAN space for image editing, these methods should be cited and discussed.

[a]  GANSpace: Discovering Interpretable GAN Controls.
[b] A Compact and Semantic Latent Space for Disentangled and Controllable Image Editing.

**Questions:**

- The motivation of this paper needs to be better explained.
- The proposed method's technical novelty could be improved.
- Editing the prompts with LLMs would be interesting to investigate.

---

### Meta-Review · Area_Chair_4CKk · 2024-12-17

**Metareview:**

This paper aims to study a new task which only allows editing a specific concept without training the text-to-image model. The authors propose an image editioning method to provide controllable image generation for specific user groups or uses cases. Experiments are performed to evaluate the effectiveness of the proposed method. Four reviewers raised concerns about the task setup, related work, novelty and presentation and they gave negative rating scores. The authors did not provide responses during the rebuttal period. Based on the above considerations, I do not recommend to accept this manuscript.

**Additional Comments On Reviewer Discussion:**

The authors did not provide responses during the rebuttal period.

---

### Decision · Program_Chairs · 2025-01-22

Reject